# No-Regret M♮-Concave Function Maximization: Stochastic Bandit Algorithms and NP-Hardness of Adversarial Full-Information Setting

**Taihei Oki**[*]
Hokkaido University
Hokkaido, Japan
oki@icredd.hokudai.ac.jp

**Shinsaku Sakaue**[*]
The University of Tokyo and RIKEN AIP
Tokyo, Japan
sakaue@mist.i.u-tokyo.ac.jp

## Abstract

M♮-concave functions, a.k.a. gross substitute valuation functions, play a fundamental role in many fields, including discrete mathematics and economics. In practice, perfect knowledge of M♮-concave functions is often unavailable a priori, and we can optimize them only interactively based on some feedback. Motivated by such situations, we study online M♮-concave function maximization problems, which are interactive versions of the problem studied by Murota and Shioura (1999). For the stochastic bandit setting, we present $O(T^{-1/2})$-simple regret and $O(T^{2/3})$-regret algorithms under $T$ times access to unbiased noisy value oracles of M♮-concave functions. A key to proving these results is the robustness of the greedy algorithm to local errors in M♮-concave function maximization, which is one of our main technical results. While we obtain those positive results for the stochastic setting, another main result of our work is an impossibility in the adversarial setting. We prove that, even with full-information feedback, no algorithms that run in polynomial time per round can achieve $O(T^{1-c})$ regret for any constant $c > 0$ unless P = NP. Our proof is based on a reduction from the matroid intersection problem for three matroids, which would be a novel idea in the context of online learning.

## 1 Introduction

M♮-concave functions form a fundamental function class in *discrete convex analysis* [32], and various combinatorial optimization problems are written as M♮-concave function maximization. In economics, M♮-concave functions (restricted to the unit-hypercube) are known as *gross substitute valuations* [19, 13, 25]; in operations research, M♮-concave functions are often used in modeling resource allocation problems [46, 30]. Furthermore, M♮-concave functions form a theoretically interesting special case of (DR-)submodular functions that the greedy algorithm can *exactly* maximize (see, Murota and Shioura [33], Murota [32, Note 6.21], and Soma [48, Remark 3.3.1]), while it is impossible for the submodular case [34, 10] and the greedy algorithm can find only *approximately* optimal solutions [35]. Due to the wide-ranging applications and theoretical importance, efficient methods for maximizing M♮-concave functions have been extensively studied [33, 46, 30, 20, 39].

When it comes to maximizing M♮-concave functions in practice, we hardly have perfect knowledge of objective functions in advance. For example, it is difficult to know the exact utility an agent gains from some items, which is often modeled by a gross substitute valuation function. Similar issues are also prevalent in submodular function maximization, and researchers have addressed them by developing *no-approximate-regret* algorithms in various settings, including stochastic/adversarial environments

---

[*]Equal contribution, alphabetical order.

38th Conference on Neural Information Processing Systems (NeurIPS 2024).

and full-information/bandit feedback [49, 14, 43, 54, 16, 36, 37, 38, 51, 55, 11, 40]. On the other hand, no-regret algorithms for M$^\natural$-concave function maximization have not been well studied, despite the aforementioned importance and practical relevance. Since the greedy algorithm can exactly solve M$^\natural$-concave function maximization, an interesting question is whether we can develop no-regret algorithms—in the standard sense *without approximation*—for M$^\natural$-concave function maximization.

## 1.1  Our contribution

This paper studies online M$^\natural$-concave function maximization for the stochastic bandit and adversarial full-information settings. Below are details of our results.

In Section 4, we study the stochastic bandit setting, where we can only observe values of an underlying M$^\natural$-concave function perturbed by sub-Gaussian noise. We first consider the stochastic optimization setting and provide an $O(T^{-1/2})$-simple regret algorithm (Theorem 4.2), where $T$ is the number of times we can access the noisy value oracle. We then convert it into an $O(T^{2/3})$-cumulative regret algorithm (Theorem 4.3), where $T$ is the number of rounds, using the explore-then-commit technique. En route to developing these algorithms, we show that the greedy algorithm for M$^\natural$-concave function maximization is *robust to local errors* (Theorem 3.1), which is one of our main technical contributions and is proved differently from related results in submodular and M$^\natural$-concave function maximization.

In Section 5, we establish the NP-hardness of no-regret learning for the adversarial full-information setting. Specifically, Theorem 5.2 shows that unless P = NP, no algorithms that run in polynomial time in each round can achieve $\mathrm{poly}(N) \cdot T^{1-c}$ regret for any constant $c > 0$, where $\mathrm{poly}(N)$ stands for any polynomial of $N$, the per-round problem size. Our proof is based on the fact that maximizing the sum of three M$^\natural$-concave functions is at least as hard as the *matroid intersection* problem for three matroids, which is known to be NP-hard.[2] We carefully construct a concrete online M$^\natural$-concave function maximization instance that enables reduction from this NP-hard problem. Our high-level idea, namely, connecting sequential decision-making and finding a common base of three matroids, might be useful for proving hardness results in other online combinatorial optimization problems.

## 1.2  Related work

There is a large stream of research on no-regret submodular function maximization. Our stochastic bandit algorithms are inspired by a line of work on explore-then-commit algorithms for stochastic bandit problems [37, 38, 11] and by a robustness analysis for extending the offline greedy algorithm to the online setting [36]. However, unlike existing results for the submodular case, the guarantees of our algorithms in Section 4 involve no approximation factors. Moreover, while robustness properties similar to Theorem 3.1 are widely recognized in the submodular case, our proof for the M$^\natural$-concave case substantially differs from them. See Appendix A for a detailed discussion.

Combinatorial bandits with linear reward functions have been widely studied [6, 9, 8, 41], and many studies have also considered non-linear functions [7, 21, 15, 29]. However, the case of M$^\natural$-concave functions has not been well studied. Zhang et al. [53] studied stochastic minimization of L$^\natural$-*convex* functions, which form another important class in discrete convex analysis [32] but fundamentally differ from M$^\natural$-convex functions. Apart from online learning, a body of work has studied maximizing valuation functions approximately from samples to do with imperfect information [2, 3, 4].

Regarding hardness results in online learning, most arguments are typically information-theoretic. For instance, the minimax regret of hopeless games in partial monitoring is $\Omega(T)$ [24, Section 37.2]. By contrast, we establish the *NP-hardness* of the adversarial full-information online M$^\natural$-concave function maximization, even though the offline M$^\natural$-concave function maximization is solvable in polynomial time. Such a situation is rare in online learning. One exception is the case studied by Bampis et al. [5]. They showed that no polynomial-time algorithm can achieve sub-linear approximate regret for some online min-max discrete optimization problems unless NP = RP, even though their offline counterparts are solvable in polynomial time. Despite the similarity in the situations, the problem class and proof techniques are completely different. Indeed, while their proof is based on the NP-hardness of determining the minimum size of a feasible solution, it can be done in polynomial time for M$^\natural$-concave function maximization [47, Corollary 4.2]. They also proved the NP-hardness of

---

[2]Note that this fact alone does not immediately imply the NP-hardness of no-regret learning since the learner can take different actions across rounds and each M$^\natural$-concave function maximization instance is *not* NP-hard.

the *multi-instance* setting, which is similar to the maximization of the sum of M$^\natural$-concave functions. However, they did not relate the hardness of multi-instance problems to that of no-regret learning.

## 2 Preliminaries

Let $V = \{1, \ldots, N\}$ be a ground set of size $N$. Let $\mathbf{0}$ be the all-zero vector. For $i \in V$, let $e_i \in \mathbb{R}^V$ denote the $i$th standard vector, i.e., the $i$th element is 1 and the others are 0; let $e_0 = \mathbf{0}$ for convenience. For $x \in \mathbb{R}^V$ and $S \subseteq V$, let $x(S) = \sum_{i \in S} x_i$. Slightly abusing notation, let $x(i) = x(\{i\}) = x_i$. For a function $f : \mathbb{Z}^V \to \mathbb{R} \cup \{-\infty\}$ on the integer lattice $\mathbb{Z}^V$, its *effective domain* is defined as $\mathrm{dom}\, f := \{ x \in \mathbb{Z}^V : f(x) > -\infty \}$. A function $f$ is called *proper* if $\mathrm{dom}\, f \neq \emptyset$. We say a proper function $f : \mathbb{Z}^V \to \mathbb{R} \cup \{-\infty\}$ is *M$^\natural$-concave* if for every $x, y \in \mathrm{dom}\, f$ and $i \in V$ with $x(i) > y(i)$, there exists $j \in V \cup \{0\}$ with $x(j) < y(j)$ or $j = 0$ such that the following inequality holds:

$$f(x) + f(y) \leq f(x - e_i + e_j) + f(y + e_i - e_j). \tag{1}$$

Similarly, we say $f : \mathbb{Z}^V \to \mathbb{R} \cup \{+\infty\}$ is *M$^\natural$-convex* if $-f$ is M$^\natural$-concave. If $x(V) \leq y(V)$, M$^\natural$-concave functions satisfy more detailed conditions, as follows.

**Proposition 2.1** (Corollary of Murota and Shioura [33, Theorem 4.2])**.** *Let $f : \mathbb{Z}^V \to \mathbb{R} \cup \{-\infty\}$ be an M$^\natural$-concave function. Then, the following conditions hold for every $x, y \in \mathrm{dom}\, f$:*
*(a) if $x(V) < y(V)$, $\exists j \in V$ with $x(j) < y(j)$, $f(x) + f(y) \leq f(x + e_j) + f(y - e_j)$ holds.*
*(b) if $x(V) \leq y(V)$, $\forall i \in V$ with $x(i) > y(i)$, $\exists j \in V$ with $x(j) < y(j)$, (1) holds.*

Let $[a, b] = \{ x \in \mathbb{Z}^V : a(i) \leq x(i) \leq b(i) \}$ be an *integer interval* of $a, b \in (\mathbb{Z} \cup \{\pm\infty\})^V$ and $f$ be M$^\natural$-concave. If $\mathrm{dom}\, f \cap [a, b] \neq \emptyset$, restricting $\mathrm{dom}\, f$ to $[a, b]$ preserves the M$^\natural$-concavity [32, Proposition 6.14]. The sum of M$^\natural$-concave functions is *not* necessarily M$^\natural$-concave [32, Note 6.16]. In this paper, we do *not* assume monotonicity, i.e., $x \leq y$ (element-wise) does not imply $f(x) \leq f(y)$.

### 2.1 Examples of M$^\natural$-concave functions

**Maximum-flow on bipartite graphs.** Let $(V, W; E)$ be a bipartite graph, where the set $V$ of $N$ left-hand-side vertices is a ground set. Each edge $ij \in E$ is associated with a weight $w_{ij} \in \mathbb{R}$. Given sources $x \in \mathbb{Z}_{\geq 0}^V$ allocated to the vertices in $V$, let $f(x)$ be the maximum-flow value, i.e.,

$$f(x) = \max_{\xi \in \mathbb{Z}_{\geq 0}^E, \, y \in \mathbb{Z}_{\geq 0}^W} \left\{ \sum_{ij \in E} w_{ij} \xi_{ij} : \forall i \in V, \sum_{j: ij \in E} \xi_{ij} = x_i; \forall j \in W, \sum_{i: ij \in E} \xi_{ij} = y_j \right\}.$$

This function $f$ is M$^\natural$-concave; indeed, more general functions specified by convex-cost flow problems on networks are M$^\natural$-concave [32, Theorem 9.27]. If we restrict the domain to $\{0, 1\}^V$ and regard $V$ as a set of items, $W$ as a set of agents, and $w_{ij} \geq 0$ as the utility of matching an item $i$ with an agent $j$, the resulting set function $f : \{0, 1\}^V \to \mathbb{R}_{\geq 0}$ coincides with the *OXS* valuation function known in combinatorial auctions [45, 25], which is a special case of the following gross substitute valuation.

**Gross substitute valuation.** In economics, an agent's valuation (a non-negative monotone set function of items) is said to be *gross substitute* (GS) if, whenever the prices of some items increase while the prices of the other items remain the same, the agent keeps demanding the same-priced items that were demanded before the price change [19, 25]. M$^\natural$-concave functions can be viewed as an extension of GS valuations to the integer lattice [32, Section 6.8]. Indeed, the class of M$^\natural$-concave functions restricted to $\{0, 1\}^V$ is equivalent to the class of GS valuations [13].

**Resource allocation.** M$^\natural$-concave functions also arise in resource allocation problems [46, 30], which are extensively studied in the operations research community. For example, given $n$ univariate concave functions $f_i : \mathbb{Z} \to \mathbb{R} \cup \{-\infty\}$ and a positive integer $K$, a function $f$ defined by $f(x) = \sum_{i=1}^n f_i(x(i))$ if $x \geq \mathbf{0}$ and $x(V) \leq K$ and $f(x) = -\infty$ otherwise is M$^\natural$-concave. More general examples of M$^\natural$-concave functions used in resource allocation are given in, e.g., Moriguchi et al. [30].

More examples can be found in Murota and Shioura [33, Section 2] and Murota [32, Section 6.3]. As shown above, M$^\natural$-concave functions are ubiquitous in various fields. However, those are often difficult to know perfectly in advance: we may neither know all edge weights in maximum-flow problems, exact valuations of agents, nor $f_i$s' values at all points in resource allocation. Such situations motivate us to study how to maximize them interactively by selecting solutions and observing some feedback.

---

**Algorithm 1** Greedy-style procedure with possibly erroneous local updates

---
1: $x_0 = \mathbf{0}$
2: **for** $k = 1, \ldots, K$ :
3:      Select $i_k \in V \cup \{0\}$            ▷ Standard greedy selects $i_k \in \arg\max_{i \in V \cup \{0\}} f(x_{k-1} + e_i)$.
4:      $x_k \leftarrow x_{k-1} + e_{i_k}$

---

### 2.2  Basic setup

Similar to bandit convex optimization [23], we consider a learner who interacts with a sequence of $M^\natural$-concave functions, $f^1, \ldots, f^T$, over $T$ rounds. To avoid incurring $f^t(x) = -\infty$, we assume that $\operatorname{dom} f^2, \ldots, \operatorname{dom} f^T$ are identical to $\operatorname{dom} f^1$. We also assume $\mathbf{0} \in \operatorname{dom} f^1$ and $\operatorname{dom} f^1 \subseteq \mathbb{Z}_{\geq 0}^V$, which are reasonable in all the examples in Section 2.1. We consider a constrained setting where the learner's action $x \in \operatorname{dom} f^1$ must satisfy $x(V) \leq K$. If $\operatorname{dom} f^1 \subseteq \{0,1\}^V$, this is equivalent to the cardinality constraint common in set function maximization. Let $\mathcal{X} := \{\, x \in \operatorname{dom} f^1 \,:\, x(V) \leq K \,\}$ denote the set of feasible actions, which the learner is told in advance. (More precisely, a $\operatorname{poly}(N)$-time membership oracle of $\mathcal{X}$ is given.) Additional problem settings specific to stochastic bandit and adversarial full-information cases are provided in Sections 4 and 5, respectively.

## 3  Robustness of greedy $M^\natural$-concave function maximization to local errors

This section studies a greedy-style procedure with possibly erroneous local updates for $M^\natural$-concave function maximization, which will be useful for developing stochastic bandit algorithms in Section 4. Let $f : \mathbb{Z}^V \to \mathbb{R} \cup \{-\infty\}$ be an $M^\natural$-concave function such that $\mathbf{0} \in \operatorname{dom} f \subseteq \mathbb{Z}_{\geq 0}^V$, which we want to maximize under $x(V) \leq K$. Let $x^* \in \arg\max\{\, f(x) \,:\, x \in \operatorname{dom} f,\, x(V) \leq K \,\}$ be an optimal solution. We consider the procedure in Algorithm 1. If $f$ is known a priori and $i_1, \ldots, i_K$ are selected as in the comment in Step 3, it coincides with the standard greedy algorithm for $M^\natural$-concave function maximization and returns an optimal solution [33]. However, when $f$ is unknown, we may select different $i_1, \ldots, i_K$ than those selected by the exact greedy algorithm. Given any $x \in \mathbb{Z}^V$ and update direction $i \in V \cup \{0\}$, we define the *local error* of $i$ at $x$ as

$$\operatorname{err}(i \,|\, x) := \max_{i' \in V \cup \{0\}} f(x + e_{i'}) - f(x + e_i) \geq 0, \tag{2}$$

which quantifies how much direction $i$ deviates from the choice of the exact greedy algorithm when $x$ is given. The following result states that local errors affect the eventual suboptimality only additively, ensuring that Algorithm 1 applied to $M^\natural$-concave function maximization is robust to local errors.

**Theorem 3.1.** *For any $i_1, \ldots, i_K \in V \cup \{0\}$, it holds that $f(x_K) \geq f(x^*) - \sum_{k=1}^{K} \operatorname{err}(i_k \,|\, x_{k-1})$.*

*Proof.* The claim is vacuously true if $\operatorname{err}(i_k \,|\, x_{k-1}) = +\infty$ occurs for some $k \leq K$. Below, we focus on the case with finite local errors. For $k = 0, 1, \ldots, K$, we define

$$\mathcal{Y}_k := \{\, y \in \mathcal{X} \,:\, y \geq x_k,\, y(V) \leq K - k + x_k(V) \,\},$$

where $y \geq x_k$ is read element-wise. That is, $\mathcal{Y}_k \subseteq \mathcal{X}$ consists of feasible points that can be reached from $x_k$ by the remaining $K - k$ updates (see Figure 1). Note that $x^* \in \mathcal{Y}_0$ and $\mathcal{Y}_K = \{x_K\}$ hold.

To prove the theorem, we will show that the following inequality holds for any $k \in \{1, \ldots, K\}$:

$$\max_{y \in \mathcal{Y}_k} f(y) \geq \max_{y \in \mathcal{Y}_{k-1}} f(y) - \operatorname{err}(i_k \,|\, x_{k-1}). \tag{3}$$

Take $y_{k-1} \in \arg\max_{y \in \mathcal{Y}_{k-1}} f(y)$ and $y_k \in \arg\max_{y \in \mathcal{Y}_k} f(y)$. If $f(y_k) \geq f(y_{k-1})$, we are done since $\operatorname{err}(i_k \,|\, x_{k-1}) \geq 0$. Thus, we assume $f(y_k) < f(y_{k-1})$, which implies $y_{k-1} \in \mathcal{Y}_{k-1} \setminus \mathcal{Y}_k$. Then, we can prove the following helper claim by using the $M^\natural$-concavity of $f$.

---

**Helper claim.** If $y_{k-1} \in \mathcal{Y}_{k-1} \setminus \mathcal{Y}_k$, there exists $j \in V \cup \{0\}$ such that $y_{k-1} + e_{i_k} - e_j \in \mathcal{Y}_k$ and

$$f(x_k) + f(y_{k-1}) \leq f(x_k - e_{i_k} + e_j) + f(y_{k-1} + e_{i_k} - e_j). \tag{4}$$

---

Assuming the helper claim, we can easily obtain (3). Specifically, (i) $f(y_{k-1} + e_{i_k} - e_j) \leq f(y_k)$ holds due to $y_{k-1} + e_{i_k} - e_j \in \mathcal{Y}_k$ and the choice of $y_k$, and (ii) $\operatorname{err}(i_k \,|\, x_{k-1}) \geq f(x_{k-1} + e_j) -$

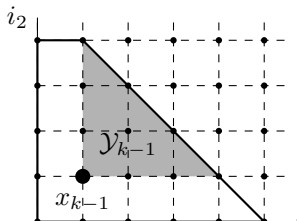 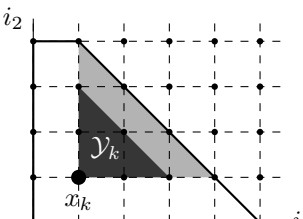 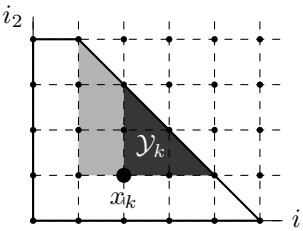

Figure 1: Images of $\mathcal{Y}_k$ on $\mathbb{Z}^2$. The set of integer points in the trapezoid is the feasible region $\mathcal{X}$. Left: the gray area represents $\mathcal{Y}_{k-1}$ consisting of points reachable from $x_{k-1}$. Middle: if $i_k = 0$ (case 1), $x_{k-1} = x_k$ holds and $\mathcal{Y}_{k-1}$ shrinks to $\mathcal{Y}_k$, the darker area, since the constraint on $y(V)$ gets tighter. Right: if $i_k = i_1$ (cases 2 and 3), the area, $\mathcal{Y}_k$, reachable from $x_k = x_{k-1} + e_{i_1}$ shifts along $e_{i_1}$.

$f(x_{k-1}+e_{i_k}) = f(x_k - e_{i_k} + e_j) - f(x_k)$ holds due to the definition of the local error (2). Combining these with (4) and rearranging terms imply (3) as follows:

$$f(y_k) \overset{\text{(i)}}{\geq} f(y_{k-1} + e_{i_k} - e_j) \overset{\text{(4)}}{\geq} f(y_{k-1}) + f(x_k) - f(x_k - e_{i_k} + e_j) \overset{\text{(ii)}}{\geq} f(y_{k-1}) - \mathrm{err}(i_k \,|\, x_{k-1}).$$

Given (3), the theorem follows from a simple induction on $k = 1, \ldots, K$. For each $k$, we will prove

$$\max_{y \in \mathcal{Y}_k} f(y) \geq f(x^*) - \sum_{k'=1}^{k} \mathrm{err}(i_{k'} \,|\, x_{k'-1}). \tag{5}$$

The case of $k = 1$ follows from (3) since $x^* \in \mathcal{Y}_0$. If it is true for $k - 1$, (3) and the induction hypothesis imply $\max_{y \in \mathcal{Y}_k} f(y) \geq \max_{y \in \mathcal{Y}_{k-1}} f(y) - \mathrm{err}(i_k \,|\, x_{k-1}) \geq f(x^*) - \sum_{k'=1}^{k-1} \mathrm{err}(i_{k'} \,|\, x_{k'-1}) - \mathrm{err}(i_k \,|\, x_{k-1})$, thus obtaining (5). Since $\mathcal{Y}_K = \{x_K\}$ holds, setting $k = K$ in (5) yields Theorem 3.1.

The rest of the proof is dedicated to proving the helper claim, which we do by examining the following three cases. The middle (right) image in Figure 1 illustrates case 1 (cases 2 and 3).

**Case 1: $i_k = 0$.** Due to $x_{k-1} = x_k$, $y_{k-1} \in \mathcal{Y}_{k-1} \setminus \mathcal{Y}_k$ implies $y_{k-1}(V) = K - (k-1) + x_{k-1}(V)$. Thus, $x_k(V) = x_{k-1}(V) = y_{k-1}(V) - (K-k+1) < y_{k-1}(V)$ holds. From Proposition 2.1 (a), there exists $j \in V$ with $x_k(j) < y_{k-1}(j)$ that satisfies (4). Also, $y_{k-1} \geq x_{k-1} = x_k$, $x_k(j) < y_{k-1}(j)$, and $(y_{k-1} - e_j)(V) = K - k + x_{k-1}(V) = K - k + x_k(V)$ imply $y_{k-1} - e_j \in \mathcal{Y}_k$.

**Case 2: $i_k \neq 0$ and $x_k(V) \leq y_{k-1}(V)$.** In this case, $y_{k-1} \in \mathcal{Y}_{k-1} \setminus \mathcal{Y}_k$ implies $y_{k-1} \geq x_{k-1}$ and $y_{k-1} \not\geq x_k = x_{k-1} + e_{i_k}$, hence $x_k(i_k) > y_{k-1}(i_k)$. From Proposition 2.1 (b), there exists $j \in V$ with $x_k(j) < y_{k-1}(j)$ that satisfies (4). Since $y_{k-1} \geq x_{k-1} = x_k - e_{i_k}$ and $x_k(j) < y_{k-1}(j)$, we have $y_{k-1} + e_{i_k} - e_j \geq x_k$. Also, we have $(y_{k-1} + e_{i_k} - e_j)(V) = y_{k-1}(V) \leq K - (k-1) + x_{k-1}(V) = K - k + x_k(V)$. Therefore, we have $y_{k-1} + e_{i_k} - e_j \in \mathcal{Y}_k$.

**Case 3: $i_k \neq 0$ and $x_k(V) > y_{k-1}(V)$.** Since $y_{k-1} \in \mathcal{Y}_{k-1}$, we have $y_{k-1} \geq x_{k-1}$. We also have $y_{k-1}(V) < x_k(V) = x_{k-1}(V) + 1$. These imply $y_{k-1} = x_{k-1}$. Therefore, (4) with $j = 0$ holds by equality, and $y_{k-1} + e_{i_k} = x_k \in \mathcal{Y}_k$ also holds. $\qquad\square$

The robustness property in Theorem 3.1 plays a crucial role in developing stochastic bandit algorithms in Section 4. Furthermore, the robustness would be beneficial beyond the application to stochastic bandits since $M^\natural$-concave functions often involve uncertainty in practice, as discussed in Section 2.1. Note that Theorem 3.1 has not been known even in the field of discrete convex analysis and that the above proof substantially differs from the original proof for the greedy algorithm *without local errors* for $M^\natural$-concave function maximization [33]. Indeed, the original proof does not consider a set like $\mathcal{Y}_k$, which is crucial in our proof. It is also worth noting that Theorem 3.1 automatically implies the original result on the errorless case by setting $\mathrm{err}(i_k \,|\, x_{k-1}) = 0$ for all $k$. We also emphasize that while Theorem 3.1 resembles robustness properties known in the submodular case [49, 14, 36, 37, 38, 11], ours is different from them in that it involves no approximation factors and requires careful inspection of the solution space, as in Figure 1. See Appendix A for a detailed discussion.

# 4 Stochastic bandit algorithms

This section presents no-regret algorithms for the following stochastic bandit setting.

**Problem setting.** For $t = 1, \ldots, T$, the learner selects $x^t \in \mathcal{X}$ and observes $f^t(x^t) = f^*(x^t) + \varepsilon^t$, where $f^* : \mathbb{Z}^V \to [0, 1] \cup \{-\infty\}$ is an unknown $M^\natural$-concave function and $(\varepsilon^t)_{t=1}^T$ is a sequence of i.i.d. 1-sub-Gaussian noises, i.e., $\mathbb{E}[\exp(\lambda \varepsilon^t)] \leq \exp(\lambda^2/2)$ for $\lambda \in \mathbb{R}$.[3] Let $x^* \in \arg\max_{x \in \mathcal{X}} f^*(x)$ denote the offline best action. In Theorem 4.2, we will discuss the simple-regret minimization setting, where the learner selects $x^{T+1} \in \mathcal{X}$ after the $T$th round to minimize the expected simple regret:

$$\mathrm{sReg}_T := f^*(x^*) - \mathbb{E}[f^*(x^{T+1})].$$

Here, the expectation is about the learner's randomness, which may originate from noisy observations and possible randomness in their strategy. This is a stochastic bandit optimization setting, where the learner aims to find the best action without caring about the cumulative regret over the $T$ rounds. On the other hand, Theorem 4.3 is about the standard regret minimization setting, where the learner aims to minimize the expected cumulative regret (or the pseudo-regret, strictly speaking):

$$\mathrm{Reg}_T := T \cdot f^*(x^*) - \mathbb{E}\left[\sum_{t=1}^T f^*(x^t)\right].$$

In this section, we assume that $T$ is large enough to satisfy $T \geq K(N + 2)$ to simplify the discussion.

**Pure-exploration algorithm.** Below, we will use a UCB-type algorithm for pure exploration in the standard stochastic multi-armed bandit problem as a building block. The algorithm is based on *MOSS* (Minimax Optimal Strategy in the Stochastic case) and is known to achieve an $O(T^{-1/2})$ simple regret as follows. For completeness, we provide the proof and the pseudo-code in Appendix B.

**Proposition 4.1** (Lattimore and Szepesvári [24, Corollary 33.3]). *Consider a stochastic multi-armed bandit instance with $M$ arms and $T'$ rounds, where $T' \geq M$. Assume that the reward of the $i$th arm in the $t$th round, denoted by $Y_i^t$, satisfies the following conditions: $\mu_i := \mathbb{E}[Y_i^t] \in [0, 1]$ and $Y_i^t - \mu_i$ is 1-sub-Gaussian. Then, there is an algorithm that, after pulling arms $T'$ times, randomly returns $i \in \{1, \ldots, M\}$ with $\mu^* - \mathbb{E}[\mu_i] = O(\sqrt{M/T'})$, where $\mu^* := \max\{\mu_1, \ldots, \mu_M\}$.*

Given this fact and our robustness result in Theorem 3.1, it is not difficult to develop an $O(T^{-1/2})$-simple regret algorithm; we select $i_k$ in Algorithm 1 with the algorithm in Proposition 4.1 consuming $\lfloor T/K \rfloor$ rounds and bound the simple regret by using Theorem 3.1, as detailed below. Also, given the $O(T^{-1/2})$-simple regret algorithm, an $O(T^{2/3})$-regret algorithm follows from the explore-then-commit technique, as described subsequently. Therefore, we think of these no-regret algorithms as byproducts and the robustness result in Theorem 3.1 as our main technical contribution on the positive side. Nevertheless, we believe those algorithms are beneficial since no-regret maximization of $M^\natural$-concave functions have not been well studied, despite their ubiquity as discussed in Section 2.1.

The following theorem presents our result regarding an $O(T^{-1/2})$-simple regret algorithm.

**Theorem 4.2.** *There is an algorithm that makes up to $T$ queries to the noisy value oracle of $f^*$ and outputs $x^{T+1}$ such that $\mathrm{sReg}_T = O(K^{3/2}\sqrt{N/T})$.*

*Proof.* Based on Algorithm 1, we consider a randomized algorithm consisting of $K$ phases. Fixing a realization of $x_{k-1}$, we discuss the $k$th phase. We consider the following multi-armed bandit instance with at most $N + 1$ arms and $\lfloor T/K \rfloor$ rounds. The arm set is $\{ i \in V \cup \{0\} : x_{k-1} + e_i \in \mathcal{X} \}$, i.e., the collection of all feasible update directions; note that the learner can construct this arm set since $\mathcal{X}$ is told in advance. In each round $t \in \{1, \ldots, \lfloor T/K \rfloor\}$, the reward of an arm $i \in V \cup \{0\}$ is given by $Y_i^t = f^*(x_{k-1} + e_i) + \varepsilon^t$, where $\varepsilon^t$ is the 1-sub-Gaussian noise. Let $\mu_i = \mathbb{E}[Y_i^t] = f^*(x_{k-1} + e_i) \in [0, 1]$ denote the mean reward of the arm $i$ and $\mu^* = \max_{i \in V \cup \{0\}} \mu_i$ the optimal mean reward. If we apply the algorithm in Proposition 4.1 to this bandit instance, it randomly returns $i_k \in V \cup \{0\}$ such that $\mathbb{E}[\mathrm{err}(i_k \mid x_{k-1}) \mid x_{k-1}] = \mu^* - \mathbb{E}[\mu_{i_k} \mid x_{k-1}] = O(\sqrt{KN/T})$, consuming $\lfloor T/K \rfloor$ queries.

---

[3] Restricting the range of $f^*$ to $[0, 1]$ and the sub-Gaussian constant to 1 is for simplicity; our results extend to any range and sub-Gaussian constant. Note that any zero-mean random variable in $[-1, +1]$ is 1-sub-Gaussian.

Consider sequentially selecting $i_k$ as above and setting $x_k = x_{k-1} + e_{i_k}$, thus obtaining $x_1, \ldots, x_K$. For any realization of $i_1, \ldots, i_K$, Theorem 3.1 guarantees $f^*(x_K) \geq f^*(x^*) - \sum_{k=1}^{K} \mathrm{err}(i_k \mid x_{k-1})$. By taking the expectations of both sides and using the law of total expectation, we obtain

$$f^*(x^*) - \mathbb{E}[f^*(x_K)] \leq \mathbb{E}\left[\sum_{k=1}^{K} \mathrm{err}(i_k \mid x_{k-1})\right] = \mathbb{E}\left[\sum_{k=1}^{K} \mathbb{E}[\mathrm{err}(i_k \mid x_{k-1}) \mid x_{k-1}]\right] = O(K^{\frac{3}{2}}\sqrt{NT}).$$

Thus, $x^{T+1} = x_K$ achieves the desired bound. The number of total queries is $K\lfloor T/K \rfloor \leq T$. $\quad\square$

We then convert the $O(T^{-1/2})$-simple regret algorithm into an $O(T^{2/3})$-regret algorithm by using the explore-then-commit technique as follows.

**Theorem 4.3.** *There is an algorithm that achieves* $\mathrm{Reg}_T = O(KN^{1/3}T^{2/3})$.

*Proof.* Let $\tilde{T} \leq T$ be the number of exploration rounds, which we will tune later. If we use the algorithm of Theorem 4.2 allowing $\tilde{T}$ queries, we can find $x^{\tilde{T}+1} \in \mathcal{X}$ with $\mathrm{sReg}_{\tilde{T}} = O(K^{3/2}\sqrt{N/\tilde{T}})$. If we commit to $x^{\tilde{T}+1}$ in the remaining $T - \tilde{T}$ rounds, the total expected regret is

$$\mathrm{Reg}_T = \mathbb{E}\left[\sum_{t=1}^{\tilde{T}} f^*(x^*) - f^*(x^t)\right] + (T - \tilde{T}) \cdot \mathrm{sReg}_{\tilde{T}} \leq \tilde{T} + T \cdot \mathrm{sReg}_{\tilde{T}} = O(\tilde{T} + TK^{\frac{3}{2}}\sqrt{N/\tilde{T}}).$$

By setting $\tilde{T} = \Theta(KN^{1/3}T^{2/3})$, we obtain $\mathrm{Reg}_T = O(KN^{1/3}T^{2/3})$. $\quad\square$

## 5 NP-hardness of adversarial full-information setting

This section discusses the NP-hardness of the following adversarial full-information setting.

**Problem setting.** An oblivious adversary chooses an arbitrary sequence of M$^\natural$-concave functions, $f^1, \ldots, f^T$, where $f^t : \mathbb{Z}^V \to [0,1] \cup \{-\infty\}$ for $t = 1, \ldots, T$, in secret from the learner. Then, for $t = 1, \ldots, T$, the learner selects $x^t \in \mathcal{X}$ and observes $f^t$, i.e., full-information feedback. More precisely, we suppose that the learner gets free access to a $\mathrm{poly}(N)$-time value oracle of $f^t$ by observing $f^t$ since M$^\natural$-concave functions may not have polynomial-size representations in general. The learner aims to minimize the expected cumulative regret:

$$\max_{x \in \mathcal{X}} \sum_{t=1}^{T} f^t(x) - \mathbb{E}\left[\sum_{t=1}^{T} f^t(x^t)\right], \tag{6}$$

where the expectation is about the learner's randomness. To simplify the subsequent discussion, we focus on the case where the constraint is specified by $K = N$ and $f^1, \ldots, f^T$ are defined on $\{0,1\}^V$; therefore, the set of feasible actions is $\mathcal{X} = \{x \in \mathrm{dom}\, f^1 : x(V) \leq K\} = \{0,1\}^V$.

For this setting, there is a simple no-regret algorithm that takes *exponential* time per round. Specifically, regarding each $x \in \mathcal{X}$ as an expert, we use the standard multiplicative weight update algorithm to select $x_1, \ldots, x_T$ [26, 12]. Since the number of experts is $|\mathcal{X}| = 2^N$, this attains an expected regret bound of $O(\sqrt{T \log |\mathcal{X}|}) \lesssim \mathrm{poly}(N)\sqrt{T}$, while taking prohibitively long $\mathrm{poly}(N)|\mathcal{X}| \gtrsim 2^N$ time per round. An interesting question is whether a similar regret bound is achievable in polynomial time per round. Thus, we focus on the *polynomial-time randomized learner*, as with Bampis et al. [5].

**Definition 5.1** (Polynomial-time randomized learner). We say an algorithm for computing $x_1, \ldots, x_T$ is a *polynomial-time randomized learner* if, given $\mathrm{poly}(N)$-time value oracles of revealed functions, it runs in $\mathrm{poly}(N, T)$ time in each round, regardless of realizations of the algorithm's randomness.[4]

Note that the per-round time complexity can depend polynomially on $T$. Thus, the algorithm can use past actions, $x_1, \ldots, x_{t-1}$, as inputs for computing $x_t$, as long as the per-round time complexity is polynomial in the input size. The following theorem is our main result on the negative side.

---

[4] While this definition does not cover so-called efficient Las Vegas algorithms, which run in polynomial time *in expectation*, requiring polynomial runtime for every realization is standard in randomized computation [1].

**Theorem 5.2.** *In the adversarial full-information setting, for any constant $c > 0$, no polynomial-time randomized learner can achieve a regret bound of $\mathrm{poly}(N) \cdot T^{1-c}$ in expectation unless $\mathsf{P} = \mathsf{NP}$.*[5]

## 5.1 Proof of Theorem 5.2

As preparation for proving the theorem, we first show that it suffices to prove the hardness for any polynomial-time *deterministic* learner and some distribution on input sequences of functions, which follows from celebrated Yao's principle [52]. We include the proof in Appendix C for completeness.

**Proposition 5.3** (Yao [52]). *Let $\mathcal{A}$ be a finite set of all possible deterministic learning algorithms that run in polynomial time per round and $\mathcal{F}^{1:T}$ a finite set of sequences of $\mathsf{M}^\natural$-concave functions, $f^1, \ldots, f^T$. Let $\mathrm{Reg}_T(a, f^{1:T})$ be the cumulative regret a deterministic learner $a \in \mathcal{A}$ achieves on a sequence $f^{1:T} = (f^1, \ldots, f^T) \in \mathcal{F}^{1:T}$. Then, for any polynomial-time randomized learner $A$ and any distribution $q$ on $\mathcal{F}^{1:T}$, it holds that*

$$\max\big\{ \mathbb{E}\big[\mathrm{Reg}_T(A, f^{1:T})\big] \ : \ f^{1:T} \in \mathcal{F}^{1:T} \big\} \geq \min\big\{ \mathbb{E}_{f^{1:T} \sim q}\big[\mathrm{Reg}_T(a, f^{1:T})\big] \ : \ a \in \mathcal{A} \big\}.$$

Note that the left-hand side is nothing but the expected cumulative regret (6) of a polynomial-time randomized learner $A$ on the worst-case input $f^{1:T}$. Thus, to prove the theorem, it suffices to show that the right-hand side, i.e., the expected regret of the best polynomial-time deterministic learner on some input distribution $q$, cannot be as small as $\mathrm{poly}(N) \cdot T^{1-c}$ unless $\mathsf{P} = \mathsf{NP}$. To this end, we will construct a finite set $\mathcal{F}^{1:T}$ of sequences of $\mathsf{M}^\natural$-concave functions and a distribution on it.

The fundamental idea behind the subsequent construction of $\mathsf{M}^\natural$-concave functions is that the matroid intersection problem for three matroids (the 3-matroid intersection problem, for short) is $\mathsf{NP}$-hard.

**3-matroid intersection problem.** A *matroid* $\mathbf{M}$ over $V$ is defined by a non-empty set family $\mathcal{B} \subseteq 2^V$ such that for $B_1, B_2 \in \mathcal{B}$ and $i \in B_1 \setminus B_2$, there exists $j \in B_2 \setminus B_1$ such that $B_1 \setminus \{i\} \cup \{j\} \in \mathcal{B}$. Elements in $\mathcal{B}$ are called *bases*. We suppose that, given a matroid, we can test whether a given $S \subseteq V$ is a base in $\mathrm{poly}(N)$ time. (This is equivalent to the standard $\mathrm{poly}(N)$-time independence testing.) The 3-matroid intersection problem asks to determine whether three given matroids $\mathbf{M}_1, \mathbf{M}_2, \mathbf{M}_3$ over a common ground set $V$ have a common base $B \in \mathcal{B}_1 \cap \mathcal{B}_2 \cap \mathcal{B}_3$ or not.

**Proposition 5.4** (cf. Schrijver [44, Chapter 41]). *The 3-matroid intersection problem is $\mathsf{NP}$-hard.*

We construct $\mathsf{M}^\natural$-concave functions that appropriately encode the 3-matroid intersection problem. Below, for any $B \subseteq V$, let $\mathbf{1}_B \in \{0, 1\}^V$ denote a vector such that $\mathbf{1}_B(i) = 1$ if and only if $i \in B$.

**Lemma 5.5.** *Let $\mathbf{M}$ be a matroid over $V$ and $\mathcal{B} \subseteq 2^V$ its base family. There is a function $f : \{0, 1\}^V \to [0, 1]$ such that (i) $f(x) = 1$ if and only if $x = \mathbf{1}_B$ for some $B \in \mathcal{B}$ and $f(x) \leq 1 - 1/N$ otherwise, (ii) $f$ is $\mathsf{M}^\natural$-concave, and (iii) $f(x)$ can be computed in $\mathrm{poly}(N)$ time at every $x \in \{0, 1\}^V$.*

*Proof.* Let $\|\cdot\|_1$ denote the $\ell_1$-norm. We construct the function $f : \{0, 1\}^V \to [0, 1]$ as follows:

$$f(x) := 1 - \frac{1}{N} \min_{B \in \mathcal{B}} \|x - \mathbf{1}_B\|_1 \quad (x \in \{0, 1\}^V).$$

Since $0 \leq \|y\|_1 \leq N$ for $y \in [-1, +1]^V$, $f(x)$ takes values in $[0, 1]$. Moreover, $\min_{B \in \mathcal{B}} \|x - \mathbf{1}_B\|_1$ is zero if $x = \mathbf{1}_B$ for some $B \in \mathcal{B}$ and at least 1 otherwise, establishing (i). Below, we show that $f$ is (ii) $\mathsf{M}^\natural$-concave and (iii) computable in $\mathrm{poly}(N)$ time.

We prove that $\tau(x) := \min_{B \in \mathcal{B}} \|x - \mathbf{1}_B\|_1$ is $\mathsf{M}^\natural$-convex, which implies the $\mathsf{M}^\natural$-concavity of $f$. Let $\delta_{\mathcal{B}} : \mathbb{Z}^V \to \{0, +\infty\}$ be the indicator function of $\mathcal{B}$, i.e., $\delta_{\mathcal{B}}(x) = 0$ if $x = \mathbf{1}_B$ for some $B \in \mathcal{B}$ and $+\infty$ otherwise. Observe that $\tau$ is the *integer infimal convolution* of $\|\cdot\|_1$ and $\delta_{\mathcal{B}}$. Here, the integer infimal convolution of two functions $f_1, f_2 : \mathbb{Z}^V \to \mathbb{R} \cup \{+\infty\}$ is a function of $x \in \mathbb{Z}^V$ defined as $(f_1 \, \square_{\mathbb{Z}} \, f_2)(x) := \min\{f_1(x - y) + f_2(y) : y \in \mathbb{Z}^V\}$, and the $\mathsf{M}^\natural$-concavity is preserved under this operation [32, Theorem 6.15]. Thus, the $\mathsf{M}^\natural$-convexity of $\tau(x) = (\|\cdot\|_1 \, \square_{\mathbb{Z}} \, \delta_{\mathcal{B}})(x)$ follows from the $\mathsf{M}^\natural$-convexity of the $\ell_1$-norm $\|\cdot\|_1$ [32, Section 6.3] and the indicator function $\delta_{\mathcal{B}}$ [32, Section 4.1].

Next, we show that $\tau(x)$ is computable in $\mathrm{poly}(N)$ time for every $x \in \{0, 1\}^V$, which implies the $\mathrm{poly}(N)$-time computability of $f(x)$. As described above, $\tau$ is the integer infimal convolution of

---

[5]Our result does not exclude the possibility of polynomial-time no-regret learning with an *exponential* factor in the regret bound. However, we believe whether $\mathrm{poly}(N) \cdot T^{1-c}$ regret is possible or not is of central interest.

$\|\cdot\|_1$ and $\delta_\mathcal{B}$, i.e., $\tau(x) = \min\{\|x-y\|_1 + \delta_\mathcal{B}(y) : y \in \mathbb{Z}^V\}$. Since the function $y \mapsto \|x-y\|_1$ is $\mathrm{M}^\natural$-convex [32, Theorem 6.15], $\tau(x)$ is the minimum value of the sum of the two $\mathrm{M}^\natural$-convex functions. While the sum of two $\mathrm{M}^\natural$-convex functions $f_1, f_2 : \mathbb{Z}^V \to \mathbb{Z}_{\geq 0} \cup \{+\infty\}$ is no longer $\mathrm{M}^\natural$-convex in general, we can minimize it via reduction to the *M-convex submodular flow* problem [32, Note 9.30]. We can solve this by querying $f_1$ and $f_2$ values $\mathrm{poly}(N, \log L, \log M)$ times, where $L$ is the minimum of the $\ell_\infty$-diameter of $\mathrm{dom}\, f_1$ and $\mathrm{dom}\, f_2$ and $M$ is an upper bound on function values [18, 17]. In our case of $f_1(y) = \|x-y\|_1$ and $f_2(y) = \delta_\mathcal{B}(y)$, we have $L = 1$ and $M \leq N$, and we can compute $f_1(y)$ and $f_2(y)$ values in $\mathrm{poly}(N)$ time (where the latter is due to the $\mathrm{poly}(N)$-time membership testing for $\mathcal{B}$). Therefore, $\tau(x)$ is computable in $\mathrm{poly}(N)$ time, and so is $f(x)$. $\qquad\square$

Now, we are ready to prove Theorem 5.2.

*Proof of Theorem 5.2.* Let $\mathbf{M}_1, \mathbf{M}_2, \mathbf{M}_3$ be three matroids over $V$ and $f_1, f_2, f_3$ functions defined as in Lemma 5.5, respectively. Let $\mathcal{F}^{1:T}$ be a finite set such that each $f^t$ $(t = 1, \ldots, T)$ is either $f_1$, $f_2$, or $f_3$. Let $q$ be a distribution on $\mathcal{F}^{1:T}$ that sets each $f^t$ to $f_1$, $f_2$, or $f_3$ with equal probability.

Suppose for contradiction that some polynomial-time deterministic learner achieves $\mathrm{poly}(N) \cdot T^{1-c}$ regret in expectation for the above distribution $q$. Let $T$ be the smallest integer such that the regret bound satisfies $\mathrm{poly}(N) \cdot T^{1-c} < \frac{T}{3N} \Leftrightarrow T > (3N\mathrm{poly}(N))^{1/c}$. Note that $T$ is polynomial in $N$ since $c > 0$ is a constant. We consider the following procedure.

> Run the polynomial-time deterministic learner on the distribution $q$ and obtain $x_t$ for $t = 1, \ldots, T$. If some $x_t$ satisfies $f_1(x_t) = f_2(x_t) = f_3(x_t) = 1$, output "Yes" and otherwise "No."

If $\mathbf{M}_1, \mathbf{M}_2, \mathbf{M}_3$ have a common base $B \in \mathcal{B}_1 \cap \mathcal{B}_2 \cap \mathcal{B}_3$, we have $f_1(\mathbf{1}_B) = f_2(\mathbf{1}_B) = f_3(\mathbf{1}_B) = 1$. On the other hand, if $x_t \neq \mathbf{1}_B$ for every $B \in \mathcal{B}_1 \cap \mathcal{B}_2 \cap \mathcal{B}_3$, $\mathbb{E}[f^t(x^t)] \leq 1 - \frac{1}{3N}$ holds from Lemma 5.5 and the fact that $f^t$ is drawn uniformly from $\{f_1, f_2, f_3\}$. Thus, to achieve the $\mathrm{poly}(N) \cdot T^{1-c}$ regret for $T > (3N\mathrm{poly}(N))^{1/c}$, the learner must return $x_t$ corresponding to some common base at least once among $T$ rounds. Consequently, the above procedure outputs "Yes." If $\mathbf{M}_1, \mathbf{M}_2, \mathbf{M}_3$ have no common base, none of $x_1, \ldots, x_T$ can be a common base, and hence the procedure outputs "No." Therefore, the above procedure returns a correct answer to the 3-matroid intersection problem. Recall that $T$ is polynomial in $N$. Since the learner runs in $T \cdot \mathrm{poly}(N, T)$ time and we can check $f_1(x_t) = f_2(x_t) = f_3(x_t) = 1$ for $t = 1, \ldots, T$ in $T \cdot \mathrm{poly}(N)$ time, the procedure runs in $\mathrm{poly}(N)$ time. This contradicts the NP-hardness of the 3-matroid intersection problem (Proposition 5.4) unless $\mathsf{P} = \mathsf{NP}$. Therefore, no polynomial-time deterministic learner can achieve $\mathrm{poly}(N) \cdot T^{1-c}$ regret in expectation. Finally, this regret lower bound applies to any polynomial-time randomized learner on the worst-case input due to Yao's principle (Proposition 5.3), completing the proof. $\qquad\square$

**Remark 5.6.** One might think that the hardness simply follows from the fact that no-regret learning in terms of (6) is too demanding. However, similar criteria are naturally met in other problems: there are efficient no-regret algorithms for online convex optimization and no-approximate-regret algorithms for online submodular function maximization. What makes online $\mathrm{M}^\natural$-concave function maximization NP-hard is its connection to the 3-matroid intersection problem, as detailed in the proof. Consequently, even though offline $\mathrm{M}^\natural$-concave function maximization is solvable in polynomial time, no polynomial-time randomized learner can achieve vanishing regret in the adversarial online setting.

## 6  Conclusion and discussion

This paper has studied no-regret $\mathrm{M}^\natural$-concave function maximization. For the stochastic bandit setting, we have developed $O(K^{3/2}\sqrt{N/T})$-simple regret and $O(KN^{1/3}T^{2/3})$-regret algorithms. A crucial ingredient is the robustness of the greedy algorithm to local errors, which we have first established for the $\mathrm{M}^\natural$-concave case. For the adversarial full-information setting, we have proved the NP-hardness of no-regret learning through a reduction from the 3-matroid intersection problem.

Our stochastic bandit algorithms are limited to the sub-Gaussian noise model, while our hardness result for the adversarial setting comes from a somewhat pessimistic analysis. Overcoming these limitations by exploring intermediate regimes between the two settings, such as stochastic bandits with adversarial corruptions [28], will be an exciting future direction from the perspective of *beyond the worst-case analysis* [42]. We also expect that our stochastic bandit algorithms have room for

improvement, considering existing regret lower bounds for stochastic combinatorial (semi-)bandits with linear reward functions. For top-$K$ combinatorial bandits, there is a sample-complexity lower bound of $\Omega(N/\varepsilon^2)$ for any $(\varepsilon, \delta)$-PAC algorithm [41]. Since our $O(K^{3/2}\sqrt{N/T})$-simple regret bound implies that we can achieve an $\varepsilon$-error in expectation with $O(K^3 N/\varepsilon^2)$ samples, our bound seems tight when $K = O(1)$, while the $K$ factors would be improvable. Regarding the cumulative regret bound, there is an $\Omega(\sqrt{KNT})$ lower bound for stochastic combinatorial semi-bandits [22]. Filling the gap between our $O(KN^{1/3}T^{2/3})$ upper bound and the lower bound is an open problem. (Since we have assumed $T = \Omega(KN)$ in Section 4, our upper bound does not contradict the lower bound.) We believe that our upper bound is essentially tight considering a recent minimax regret bound by Tajdini et al. [50] for bandit submodular maximization, which we discuss in detail below. Regarding the adversarial setting, it will be interesting to explore no-approximate-regret algorithms. If $M^{\natural}$-concave functions are restricted to $\{0, 1\}^V$, the resulting problem is a special case of online submodular function maximization and hence vanishing $1/2$-approximate regret is already possible [43, 16, 36]. We may be able to improve the approximation factor by using the $M^{\natural}$-concavity.

**Discussion on the tightness of the $O(KN^{1/3}T^{2/3})$ bound.** As mentioned above, obtaining a tight regret bound for stochastic bandit $M^{\natural}$-concave maximization is left open. Nevertheless, we conjecture that our $O(KN^{1/3}T^{2/3})$ bound in Theorem 4.3 is tight unless we admit exponential factors in $K$. The rationale behind this conjecture lies in a recent result by Tajdini et al. [50]. They studied stochastic bandit monotone submodular maximization with a ground set of size $N$ and a cardinality constraint of $K$, and they showed that there is a lower bound of

$$\Omega\left((K - i)N^{1/3}T^{2/3} + \sqrt{\binom{N - K}{i}T}\right)$$

on *robust greedy regret*, which compares the learner's actual reward with the output of the greedy algorithm, denoted by $S_{\mathrm{gr}}$, applied to the underlying true submodular function. Here, $i \le K$ is the largest positive integer with $\frac{16}{N^2 K^6}\binom{N-K}{i}^3 \le T$; see Tajdini et al. [50, Theorem 2.3] for details.[6] This lower bound suggests that the $O(KN^{1/3}T^{2/3})$ regret for stochastic bandit submodular maximization, which can also be achieved by the explore-then-commit strategy, is inevitable in general. We can interpret the $\sqrt{\binom{N-K}{i}T}$ term as the regret achieved by regrading all $\binom{N-K}{i}$ subsets as arms and using a UCB-type algorithm. Thus, the lower bound consists of the two regret terms achieved by the explore-then-commit and the UCB applied to exponentially many arms.

Currently, we have observed that the proof of the lower bound by Tajdini et al. [50] does not directly apply to our stochastic bandit $M^{\natural}$-concave maximization problem. Specifically, the function used in their proof for obtaining the lower bound is submodular but not $M^{\natural}$-concave. Nevertheless, the problem setting of Tajdini et al. [50] and our problem in Section 4, with the domain restricted to $\{0, 1\}^V$, have notable connections:

1. Since the greedy algorithm applied to the unknown true $M^{\natural}$-concave function $f^*$ can find an optimal solution $x^*$, we have $x^* = S_{\mathrm{gr}}$. Therefore, the notion of robust greedy regret in Tajdini et al. [50] essentially coincides with the standard regret in our case.

2. Both the $O(KN^{1/3}T^{2/3})$ and $O\left(\sqrt{\binom{N-K}{i}T}\right)$ regret bounds discussed above can also be achieved by the explore-then-commit and UCB strategies, respectively, in our $M^{\natural}$-concave case, where the former is exactly what our Theorem 4.3 states.

Considering these facts, we expect that we can construct a hard instance of stochastic bandit $M^{\natural}$-concave maximization similar to Tajdini et al. [50] to establish the same regret lower bound. Therefore, we conjecture that our $O(KN^{1/3}T^{2/3})$ regret bound in Theorem 4.3 is tight in $K$, $N$, and $T$, if we want to avoid the exponential factor, which generally scales as $N^K$, regardless of the value of $T$.

---

[6]More precisely, the lower bound of Tajdini et al. [50, Theorem 2.3] applies to the class of *non-adaptive* greedy algorithms, which specify error thresholds only depending on $T$, $N$, and $K$. Our algorithm in Section 4, which runs MOSS in each iteration for $\lfloor T/K \rfloor$ rounds, falls into this category. Tajdini et al. [50, Theorem 2.1] also shows that a weaker lower bound, with the first term replaced with $(K - i)^{1/3}N^{1/3}T^{2/3}$, applies to all stochastic bandit submodular maximization algorithms.

## Acknowledgements

The authors thank the anonymous reviewers for their valuable feedback, particularly for bringing our attention to the recent result by Tajdini et al. [50]. This work was supported by JST ERATO Grant Number JPMJER1903, JST CREST Grant Number JPMJCR24Q2, JST FOREST Grant Number JPMJFR232L, and JSPS KAKENHI Grant Numbers JP22K17853 and 24K21315.

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

## A    Differences of Theorem 3.1 from robustness results in the submodular case

The basic idea of analyzing the robustness is inspired by similar approaches used in online submodular function maximization [49, 14, 36, 37, 38, 11]. However, our Theorem 3.1 for the $M^\natural$-concave case is fundamentally different from those for the submodular case.

At a high level, an evident difference lies in the comparator in the guarantees. Specifically, we need to bound the suboptimality compared to the *optimal* value in the $M^\natural$-concave case, while the comparator is an *approximate* value in the submodular case.

At a more technical level, we need to work on the solution space in the $M^\natural$-concave case, while the proof for the submodular case follows from analyzing objective values directly. Let us overview the standard technique for the case of monotone submodular function maximization under the cardinality constraint, which is the most relevant to our case due to the similarity in the algorithmic procedures. In this case, a key argument is that in each iteration, the marginal increase in the objective value is lower bounded by a $1/K$ fraction of that gained by adding an optimal solution, minus the local error. That is, regarding $f : \{0, 1\}^V \to \mathbb{R}$ as a submodular set function, the submodularity implies $f(x_k) - f(x_{k-1}) \geq \frac{1}{K}(f(x_{k-1} \vee x^*) - f(x_{k-1})) - \mathrm{err}(i_k \,|\, x_{k-1})$, where $\vee$ is the element-wise maximum. Consequently, by rearranging terms in the same way as the proof of the $(1 - 1/\mathrm{e})$-approximation, one can confirm that local errors accumulate only additively over $K$ iterations. In this way, the robustness property directly follows from incorporating the effect of local errors into the inequality for deriving the $(1 - 1/\mathrm{e})$-approximation in the submodular case. By contrast, in our proof of Theorem 3.1 for the $M^\natural$-concave case, we need to look at the solution space to ensure that the local update by $i_k$ with small $\mathrm{err}(i_k \,|\, x_{k-1})$ does not deviate much from $\mathcal{Y}_{k-1}$, as highlighted in (3) (and this also differs from the original proof without errors [33]). After establishing this, we can obtain the theorem by induction by virtue of the non-trivial design of $\mathcal{Y}_k$ $(k = 0, \ldots, K)$, which satisfies $x^* \in \mathcal{Y}_0$ and $\mathcal{Y}_K = \{x_K\}$.

## B    MOSS for pure exploration in stochastic multi-armed bandit

We overview the MOSS-based pure-exploration algorithm used in Section 4. For more details, see Lattimore and Szepesvári [24, Chapters 9 and 33].

Let $\mathbb{I}\{A\}$ take 1 if $A$ is true and 0 otherwise, and let $\log^+(x) = \log \max\{1, x\}$. Given a stochastic multi-armed bandit instance with $M$ arms and $T'$ rounds, we consider an algorithm that randomly selects arms $A^1, \ldots, A^{T'} \in \{1, \ldots, M\}$. For $t = 1, \ldots, T'$, let $Y^t$ be a random variable representing the learner's reward in the $t$th round, $\hat{\tau}_i(t) = \sum_{s=1}^t \mathbb{I}\{A^s = i\}$ the number of times the $i$th arm is selected up to round $t$, and $\hat{\mu}_i(t) = \frac{1}{\hat{\tau}_i(t)} \sum_{s=1}^t \mathbb{I}\{A^s = i\} Y^s$ the empirical mean reward of the $i$th arm up to round $t$. Given these definitions, the MOSS algorithm can be described as in Algorithm 2.

---
**Algorithm 2** MOSS

**Input:** Bandit instance with $M$ arms and $T'$ rounds
 1: Choose each arm $i \in \{1, \ldots, M\}$ during the first $M$ rounds
 2: **for** $t = M + 1, \ldots, T'$ **:**

 3:     Choose $A^t = \arg\max_{i \in \{1, \ldots, M\}} \hat{\mu}_i(t-1) + \sqrt{\frac{4}{\hat{\tau}_i(t-1)} \log^+\left(\frac{T'}{N \hat{\tau}_i(t-1)}\right)}$

---

Let $a^1, \ldots, a^{T'}$ denote the realization of $A^1, \ldots, A^{T'}$, respectively, after running the MOSS algorithm. Then, we set the final output to $i \in \{1, \ldots, M\}$ with probability $\frac{1}{T'} \sum_{t=1}^{T'} \mathbb{I}\{a^t = i\}$. This procedure gives an $O(\sqrt{M/T'})$-simple regret algorithm, as stated in Proposition 4.1.

**Proposition 4.1** (Lattimore and Szepesvári [24, Corollary 33.3])**.** *Consider a stochastic multi-armed bandit instance with $M$ arms and $T'$ rounds, where $T' \geq M$. Assume that the reward of the $i$th arm in the $t$th round, denoted by $Y_i^t$, satisfies the following conditions: $\mu_i := \mathbb{E}[Y_i^t] \in [0, 1]$ and $Y_i^t - \mu_i$ is 1-sub-Gaussian. Then, there is an algorithm that, after pulling arms $T'$ times, randomly returns $i \in \{1, \ldots, M\}$ with $\mu^* - \mathbb{E}[\mu_i] = O(\sqrt{M/T'})$, where $\mu^* := \max\{\mu_1, \ldots, \mu_M\}$.*

*Proof.* Since the suboptimality of the $i$th arm, defined by $\mu^* - \mu_i$, is at most 1 for all $i \in \{1, \ldots, M\}$, the MOSS algorithm enjoys a cumulative regret bound of $\mathrm{Reg}_{T'} := T' \cdot \mu^* - \mathbb{E}\left[\sum_{t=1}^{T'} \mu_{A^t}\right] \le 39\sqrt{MT'} + M$ (see Lattimore and Szepesvári [24, Theorem 9.1]). Consider setting the final output to $i \in \{1, \ldots, M\}$ with probability $\frac{1}{T'}\sum_{t=1}^{T'} \mathbb{I}\{a^t = i\}$, where $a^t$ denote the realization of $A^t$. Then, it holds that $\mu^* - \mathbb{E}_{i \sim p}[\mu_i] = \mathrm{Reg}_{T'}/T'$ (see Lattimore and Szepesvári [24, Proposition 33.2]). The right-hand side is at most $(39\sqrt{MT'} + M)/T' \le 40\sqrt{M/T'}$, completing the proof. $\qquad\square$

## C   Proof of Proposition 5.3

**Proposition 5.3** (Yao [52])**.** *Let $\mathcal{A}$ be a finite set of all possible deterministic learning algorithms that run in polynomial time per round and $\mathcal{F}^{1:T}$ a finite set of sequences of $M^{\natural}$-concave functions, $f^1, \ldots, f^T$. Let $\mathrm{Reg}_T(a, f^{1:T})$ be the cumulative regret a deterministic learner $a \in \mathcal{A}$ achieves on a sequence $f^{1:T} = (f^1, \ldots, f^T) \in \mathcal{F}^{1:T}$. Then, for any polynomial-time randomized learner $A$ and any distribution $q$ on $\mathcal{F}^{1:T}$, it holds that*

$$\max\left\{\mathbb{E}\left[\mathrm{Reg}_T(A, f^{1:T})\right] \,:\, f^{1:T} \in \mathcal{F}^{1:T}\right\} \ge \min\left\{\mathbb{E}_{f^{1:T} \sim q}\left[\mathrm{Reg}_T(a, f^{1:T})\right] \,:\, a \in \mathcal{A}\right\}.$$

*Proof.* We use the same proof idea as that of Yao's principle (see, e.g., Motwani and Raghavan [31, Section 2.2]). First, note that any polynomial-time randomized learner can be viewed as a polynomial-time deterministic learner with access to a random tape. Thus, we can take $A$ to be chosen according to some distribution $p$ on the family, $\mathcal{A}$, of all possible polynomial-time deterministic learners.

Consider an $|\mathcal{A}| \times |\mathcal{F}^{1:T}|$ matrix $M$, whose entry corresponding to row $a \in \mathcal{A}$ and column $f^{1:T} \in \mathcal{F}^{1:T}$ is $\mathrm{Reg}_T(a, f^{1:T})$. For any polynomial-time randomized learner $A$ and any distribution $q$ on $\mathcal{F}^{1:T}$, it holds that

$$\begin{aligned}
\max\left\{\mathbb{E}\left[\mathrm{Reg}_T(A, f^{1:T})\right] \,:\, f^{1:T} \in \mathcal{F}^{1:T}\right\} &\ge \min_{p'} \max_{e_{f^{1:T}}} \; p'Me_{f^{1:T}} \\
&= \max_{q'} \min_{e_a} \; e_a Mq' \\
&\ge \min\left\{\mathbb{E}_{f^{1:T} \sim q}\left[\mathrm{Reg}_T(a, f^{1:T})\right] \,:\, a \in \mathcal{A}\right\},
\end{aligned}$$

where $p'$ and $q'$ denote probability vectors on $\mathcal{A}$ and $\mathcal{F}^{1:T}$, respectively, and $e_{f^{1:T}}$ and $e_a$ denote the standard unit vectors of $f^{1:T}$ and $a$, respectively. The equality is due to Loomis' theorem [27]. $\quad\square$

