# OpenReview forum: "No-Regret M${}^{\natural}$-Concave Function Maximization: Stochastic Bandit Algorithms and NP-Hardness of Adversarial Full-Information Setting"
_NeurIPS.cc/2024/Conference — NeurIPS 2024 poster_

### Official Review · Reviewer_PdPM · 2024-07-08

**Soundness:** 3
**Presentation:** 2
**Contribution:** 2
**Rating:** 6
**Confidence:** 4

**Summary:**

This paper considers online learning variants of ${\rm M}^\natural$-concave function maximization.  These types of function generalize maximum flows on graphs (where the variable is the vector of source values on each node), gross substitute valuations in economics, and have applications in resource allocation.

This paper has two main results:

 - A learning strategy with $O(K N^{1/3} T^{2/3})$-regret in the stochastic setting (here we can query a function at a given input and observe its output perturbed by mean 0, 1-subgaussian noise and we have $T$ rounds to perform these queries).  This is obtained by showing that the greedy algorithm is "robust to local errors", and is the key technical step for this result.  The regret bound follows by an application of known results for pure-exploration + the explore-then-commit paradigm.

 - NP-hardness for the adversarial setting.  Unless P=NP, it is impossible for any polynomial time learner to achieve sub-linear regret in the adversarial setting.  This is done via a clever reduction from the 3-matroid-intersection problem.

**Strengths:**

- This paper considers online learning for a very general class of functions, which makes the results potentially widely applicable.
 - The "robust to local errors" Theorem for $\rm{M}^\natural$-concave functions may be of independent interest.
 - The proofs are well written and easy to follow, which is challenging to do for the very technical topic at hand.
 - The NP-hardness result is interesting and shows a sharp difference between the stochastic and adversarial case for this problem.

**Weaknesses:**

- For the upper bound, the techniques are fairly straightforward once we have Theorem 3.1.  I am wondering if more can be done, e.g. regret that is sublinear in $K$, improving the bound for $T$ to $O(\sqrt{T})$, or sublinear approximate regret in the adversarial setting.  These suggestions have been pointed out by the authors themselves, but they are relevant questions.
- The stochastic online setting for this problem could be better motivated than the paragraph in lines 122-126.  More concrete examples would be helpful to clarify the applicability of these results beyond the nice mathematics.
- No experimental evaluation.  While this paper makes a nice theoretical contribution, it could be helpful to demonstrate that the theory is applicable to a problem that prior methods were not suitable for.

**Questions:**

Please clarify the relationship between ${\rm M}^{\natural}$ concave maximization and submodular function maximization.  I found it hard to understand the relationship between these as it pertains to the results in this paper and the discussion therein.  For example, the conclusion states that if the domain is restricted to $\{0,1\}^V$ then we get online submodular function maximization as a special case.  Here, the offline problem is NP-hard so I'm not sure how to interpret the fact that we can get sublinear-regret with respect to $\mathbb{E}[f^*(x^*)]$ with a polynomial-time algorithm.

Some very minor comments:
- ${\rm M}^\natural$-convexity is used in section 5, but not defined in the paper

**Limitations:**

The authors have adequately addressed and discussed limitations of their work.

---

> ### Author Rebuttal · Authors · 2024-08-05
>
> We are truly grateful to the reviewer for the thoughtful comments and positive evaluation. First, we would like to address the following comment regarding weakness.
>
> > Weaknesses:
> > - For the upper bound, the techniques are fairly straightforward once we have Theorem 3.1. I am wondering if more can be done, e.g. regret that is sublinear in $K$, improving the bound for $T$ to $O(\sqrt{T})$, or sublinear approximate regret in the adversarial setting. These suggestions have been pointed out by the authors themselves, but they are relevant questions.
>
> We appreciate the feedback and insights provided. As regards improving the $O(KN^{1/3}T^{2/3})$ regret bound in Theorem 4.3 for the stochastic bandit setting, we have discovered a recent preprint by Tajdini et al. (2023) that presents a relevant lower bound. In the submodular case, their lower bound implies that significant improvements to the $O(KN^{1/3}T^{2/3})$ regret bound are impossible without admitting exponential factors in $K$. While their result does not directly apply to our $\text{M}^\natural$-concave setting, we conjecture that a similar lower bound exists and that our $O(KN^{1/3}T^{2/3})$ regret bound is tight unless we admit exponential factors in $K$. For more details, please refer to the above [global response](https://openreview.net/forum?id=NnoAj91HZX&noteId=RjP8EjP159).
>
> Next, we would like to answer the following question.
>
> > Questions:
> >
> > Please clarify the relationship between $\text{M}^\natural$-concave maximization and submodular function maximization. I found it hard to understand the relationship between these as it pertains to the results in this paper and the discussion therein. For example, the conclusion states that if the domain is restricted to ${0, 1}^V$ then we get online submodular function maximization as a special case. Here, the offline problem is NP-hard so I'm not sure how to interpret the fact that we can get sublinear-regret with respect to $\mathbb{E}[f^*(x^*)]$ with a polynomial-time algorithm.
>
> We wish to clarify that the correct relationship we intended to describe in the conclusion is:
> $$
> \text{class of $\text{M}^\natural$-concave functions on $\\{0, 1\\}^V$}\subseteq \text{class of subomdular functions},
> $$
> which is the opposite of the relationship mentioned in the reviewer's comment. We apologize for any confusion and appreciate it if you could notify us of any incorrect expressions in our manuscript.
>
> Importantly, $\text{M}^\natural$-concave maximization on $\\{0, 1\\}^V$ is a special case that the greedy algorithm can *exactly* solve in polynomial time. This has been established by Murota and Shioura [33] and can also be derived from our Theorem 3.1 with $\mathrm{err}(i_k \mid x_{k-1}) = 0$. Since submodular maximization forms a larger problem class, sublinear regret with respect to $\mathbb{E}[f^*(x^*)]$ for stochastic bandit $\text{M}^\natural$-concave maximization does *not* imply exact algorithms for submodular maximization. Therefore, our sublinear regret bounds in Section 4 do not contradict any known results in $\text{M}^\natural$-concave or submodular maximization.
>
> We hope this clarification has effectively addressed the reviewer's concerns. Please do not hesitate to let us know if further questions remain.
>
> > Some very minor comments:
> > - $\text{M}^\natural$-convexity is used in section 5, but not defined in the paper
>
> We thank the reviewer for pointing this out. We will clarify that the $\text{M}^\natural$-convexity is defined by the negative of $\text{M}^\natural$-concavity, analogous to the standard convexity--concavity relationship.

---

> > ### Comment · Reviewer_PdPM · 2024-08-07
> >
> > Thank you for the detailed response to my specific questions and the general response above.  For the M-concave vs. submodular discussion above my issue was one of slight confusion with what is written in the text.  Your response has cleared this up.  My overall evaluation stays the same.

---

### Official Review · Reviewer_opkE · 2024-07-11

**Soundness:** 3
**Presentation:** 3
**Contribution:** 2
**Rating:** 6
**Confidence:** 3

**Summary:**

The authors consider online $M^\sharp$-concave optimization, similar to problems like online convex optimization and online DR-submodular optimization.  $M^\sharp$-concave function classes include resource allocation, valuation, and flow problems, and unlike DR-submodular functions can be exactly optimized (at least under a cardinality constraint) by a simple greedy algorithm.  The authors show standard online adaptation of the greedy algorithm analogous to similar work for submodular functions, though prove the robustness of the offline algorithm to oracle value errors in a manner distinct from previous robustness analyses (such as for submodular functions).  The authors also show that unlike OCO and online DR-submodular optimization, the adversarial setting is fundamentally harder than the stochastic setting – one cannot (with poly-time per round complexity) achieve sublinear regret.

**Strengths:**

1. The authors consider an interesting class of functions for online optimization, which are related to, but much easier to optimize than, (DR-)submodular functions.  They show an (potentially quite) interesting dichotomy in hardness between the online stochastic setting and the online adversarial setting, unlike related classes like online convex optimization or online DR-submodular optimization.  For the online stochastic setting they get analogous results to prior work for (DR-)submodular, but show through a matroid-intersection construction that one cannot obtain sublinear regret.

2. The robustness analysis is distinct from that used in submodular bandit papers.  As described, the standard proof(s) for offline cardinality-constrained $M^\sharp$-concave maximization does not have a structure that lends itself to accounting for the function value impact in sequential mistakes by the greedy algorithm.  Thus, the authors develop a new proof that clearly shows such additive accumulation, and subsequently permits almost direct adaptation to the online stochastic setting.  (one minor note - line 191 “ours is different from them in that it involves no approximation factors” that aspect I don’t see in and of itself as a meaningful distinction since the offline problem is not NP-hard; from my reading it is how the offline proof is structured as to whether it can be easily modified to account for value oracle errors that matters.)

3.  I found the paper overall well written, with good organization, logical flow, discussions, etc.

**Weaknesses:**

1. The stochastic setting algorithms and analysis adapting a greedy algorithm from the offline setting (including an ETC method getting $T^{2/3}$) are straightforward (once a robustness result is in hand), though the authors do acknowledge that in the main paper.

2. This is a somewhat minor point – since the need for a novel robustness analysis arises due to the offline proof of the greedy algorithm’s not already being in a manner that can be easily adapted to account for local errors (which is unlike the proof of $1-1/e$ for the greedy approximation algorithm for cardinality constrained submodular maximization, as noted in Appendix A), I think it would have been helpful to include the proof(s) for constrained $M^\sharp$-concave maximization (specialized to cardinality constraint) so that this need would be more apparent, that no standard proof has a sequential exchange argument that would already lend itself more readily to analyzing robustness and that the robustness result used in the offline setting with exact value oracles clearly constitutes a distinct proof.

3. minor point – it is not clear to what extent the positive result in the stochastic setting can be generalized beyond a simple cardinality constraint.

4. minor point – there is no implementation or experiments for toy flow/valuation/resource allocation problems.

**Questions:**

1. What is known about the hardness of offline optimization for maximizing sums of (possibly non-monotone) $M^\sharp$-concave functions?
For the adversarial setting, the authors use an interesting construction based on matroid intersection to prove hardness for no-approximation sublinear regret (with poly-time per round computation).  But I wonder if it could have been reached in a more straightforward manner.  Namely, for the stochastic setting, the regret is based on a single $M^\sharp$-concave function $f^*$ for which in the offline setting the greedy algorithm can find the optimal solution. In the adversarial setting, however, the regret is based on a sum of $M^\sharp$-concave functions and in line 100 we are told that in general the class of $M^\sharp$ functions is not closed under addition.

**Limitations:**

Yes

---

> ### Author Rebuttal · Authors · 2024-08-05
>
> We deeply appreciate the reviewer's dedication to reviewing our paper and providing many insightful comments.
>
> First, we would like to respond to some comments regarding weaknesses.
>
> > Weaknesses:
> >
> > 1. The stochastic setting algorithms and analysis adapting a greedy algorithm from the offline setting (including an ETC method getting $T^{-2/3}$) are straightforward (once a robustness result is in hand), though the authors do acknowledge that in the main paper.
>
> We acknowledge the reviewer's point. Indeed, while our algorithm is straightforward, significantly improving our $O(T^{2/3})$ regret seems challenging unless we admit exponential dependence on $K$. Please refer to the above [global response](https://openreview.net/forum?id=NnoAj91HZX&noteId=RjP8EjP159) for details.
>
> > 2. This is a somewhat minor point – since the need for a novel robustness analysis arises due to the offline proof of the greedy algorithm’s not already being in a manner that can be easily adapted to account for local errors (which is unlike the proof of $1-1/e$ for the greedy approximation algorithm for cardinality constrained submodular maximization, as noted in Appendix A), I think it would have been helpful to include the proof(s) for constrained $M^\sharp$-concave maximization (specialized to cardinality constraint) so that this need would be more apparent, that no standard proof has a sequential exchange argument that would already lend itself more readily to analyzing robustness and that the robustness result used in the offline setting with exact value oracles clearly constitutes a distinct proof.
>
> We appreciate this valuable suggestion. The original proof by Murota and Shioura [33 Section 3], which is based on a convexity argument along the trajectory of an augmenting-type algorithm, is a quite different approach that does not readily align with the robustness analysis. To add further, we wish to highlight that our proof of Theorem 3.1, with $\mathrm{err}(i_k \mid x_{k-1}) = 0$, serves as an alternative proof of the optimality of the greedy algorithm for the offline $\text{M}^\natural$-concave maximization.
>
> > 3. minor point – it is not clear to what extent the positive result in the stochastic setting can be generalized beyond a simple cardinality constraint.
>
> This is an interesting future direction. Indeed, in the offline setting, the greedy-type algorithm can solve more general $\text{M}^\natural$-concave maximization problems. When it comes to online/bandit settings, extending the robustness analysis akin to Theorem 3.1 will be a key point of investigation.
>
> Next, we would like to respond to the following question.
>
> > Questions:
> >
> > 1. What is known about the hardness of offline optimization for maximizing sums of (possibly non-monotone) $M^\sharp$-concave functions? For the adversarial setting, the authors use an interesting construction based on matroid intersection to prove hardness for no-approximation sublinear regret (with poly-time per round computation). But I wonder if it could have been reached in a more straightforward manner. Namely, for the stochastic setting, the regret is based on a single $M^\sharp$-concave function $f^*$ for which in the offline setting the greedy algorithm can find the optimal solution. In the adversarial setting, however, the regret is based on a sum of $M^\sharp$-concave functions and in line 100 we are told that in general the class of $M^\sharp$ functions is not closed under addition.
>
> In the offline setting, maximizing the sum of more than two $\text{M}^\natural$-concave functions is NP-hard in general. This is recognized as folklore within the field of discrete convex analysis; those familiar with $\text{M}^\natural$-concavity would readily infer this NP-hardness due to the connection to the 3-matroid intersection, by encoding a base family of a matroid with an $\text{M}^\natural$-concave indicator function $f:\\{0, 1\\}^V \to \\{0, -\infty\\}$. To our knowledge, however, no explicit proof exists in the literature (although the NP-hardness is mentioned in a seminar material by Kazuo Murota, titled "Discrete Convex Analysis," used in the 2015 Summer School at Hausdorff Institute of Mathematics). Our work would be the first to provide an explicit proof, although the NP-hardness itself is already widely recognized.
>
> More importantly, our Lemma 5.5 implies a slightly stronger result: even if $\text{M}^\natural$-concave functions on $\\{0, 1\\}^V$ are restricted to take values in $[0, 1]$ (forbidden taking $-\infty$), maximizing the sum of more than two $\text{M}^\natural$-concave functions remains NP-hard. This restriction is crucial to ensure that our NP-hardness result is meaningful because, if $f_t$ can take $-\infty$, the learner who does not know $f_t$ a priori cannot achieve even a bounded regret, making the hardness result (Theorem 5.2) vacuous. The reduction from the 3-matroid intersection with $\text{M}^\natural$-concave functions that do not take $-\infty$ has not been even recognized in the literature. Thus, our Lemma 5.5, despite being somewhat specific, provides a new crucial technical result. We believe that no significantly more straightforward approach could establish meaningful NP-hardness.
>
> We hope this clarification has effectively addressed the reviewer's question and highlighted the value of our technical contributions. Please do not hesitate to reach out during the discussion period if further questions remain.

---

> > ### Comment · Reviewer_opkE · 2024-08-13
> >
> > I have read the rebuttal.  Thanks to the authors for their response.  I have decided to increase my score.  I do not have further questions.

---

### Official Review · Reviewer_ugrH · 2024-07-13

**Soundness:** 3
**Presentation:** 3
**Contribution:** 3
**Rating:** 6
**Confidence:** 3

**Summary:**

This paper studies the online version of M-concave function maximization. With the help of a new lemma that bounds local errors for greedy algorithms, the authors derive an efficient $T^{-1/2}$ simple regret algorithm together with a $T^{2/3}$ regret algorithm for stochastic settings. Moreover, the authors show an interesting computational hardness result for adversarial settings with full information.

**Strengths:**

1. This paper studies a new online learning problem and proposes an efficient algorithm with sublinear regret for stochastic settings.
2. A new computational hardness result for adversarial settings with full information is given, which introduces a new understanding of online optimization.

**Weaknesses:**

1. The regret bound for stochastic setting is not tight.

**Questions:**

1. What is the main technical challenge in getting $\sqrt{K}$ regret for stochastic settings? Specifically, what is the difficulty when implementing the UCB-type exploration on the fly rather than doing an explore-then-commit algorithm?

**Limitations:**

Yes.

---

> ### Author Rebuttal · Authors · 2024-08-05
>
> We really appreciate the reviewer's effort in reviewing our paper and providing thoughtful comments. Below is our answer to the question.
>
> > Questions:
> >
> > 1. What is the main technical challenge in getting $\sqrt{K}$ regret for stochastic settings? Specifically, what is the difficulty when implementing the UCB-type exploration on the fly rather than doing an explore-then-commit algorithm?
>
> We appreciate this insightful question. Our response assumes the question refers to $\sqrt{T}$ regret (not $\sqrt{K}$). Please let us know if this is not your intended question!
>
> The fundamental difficulty lies in the fact that there are exponentially many arms. Indeed, if the domain is restricted to $\\{0, 1\\}^V$, we can achieve a $\sqrt{T}$ regret bound by regarding all subsets of size up to $K$ as arms and naively applying a UCB-type algorithm to this multi-armed bandit problem. However, this approach leads to a regret bound of $O\left(\sqrt{\binom{N}{K}T}\right)$, which depends exponentially on $K$, and the resulting algorithm requires exponential time per round in general. By contrast, the explore-then-commit strategy can achieve the $O(KN^{1/3}T^{2/3})$ regret bound as in Theorem 4.3, albeit with the worse dependence on $T$, and the algorithm runs in polynomial time per round.
>
> As detailed in the [global response](https://openreview.net/forum?id=NnoAj91HZX&noteId=RjP8EjP159), a recent lower bound by Tajdini et al. (2023) implies that, in the submodular maximization case, we cannot do significantly better than the naive UCB applied to $\binom{N}{K}$ arms and the explore-then-commit strategy. While there is a slight possibility of achieving better regret bounds by focusing on the $\text{M}^\natural$-concave case, we conjecture that a similar lower bound likely exists and that our $O(KN^{1/3}T^{2/3})$ regret is essentially tight unless we allow the regret bound to depend exponentially on $K$.
>
> We hope this explanation helps the reviewer better understand the challenge in stochastic bandit $\text{M}^\natural$-concave maximization and casts our results in a more favorable light. Please do not hesitate to ask further questions during the discussion period.

---

> > ### Comment · Reviewer_ugrH · 2024-08-10
> >
> > Thank you for the detailed response. Yeah, my question is about the difficulty of getting $\sqrt{T}$ regret. I do not have further questions now.

---

### Official Review · Reviewer_yBbo · 2024-07-13

**Soundness:** 3
**Presentation:** 3
**Contribution:** 3
**Rating:** 6
**Confidence:** 4

**Summary:**

This paper studies the online optimization problem with $M^\natural$-concave function, which has many real-world implications such as maximum-flow on bipartite graphs, gross substitute valuation, resource allocation and bandit problem. $M^\natural$-concave function forms the fundamental basis of discrete concave analysis. In the stochastic bandit setting, they present algorithms with $O(T^-1/2)$ simple regret and $O(T^2/3)$ cumulative regret. These results leverage the robustness of the greedy algorithm to local errors, a significant technical contribution of the paper.

Additionally, the author also presents the result in the adversarial online learning setting, the paper proves that achieving sub-linear regret is NP-hard, even with full-information feedback, which establishes a distinct difference from the offline setting.

**Strengths:**

The paper focuses on $M^\natural$-concave functions, which is a crucial class in discrete convex analysis with wide applications, thus benefiting the analysis on a large variety community.
The author brings both stochastic bandit and adversarial full-information settings.
Especially for the adversarial setting, at least linear regret justification will guide us to avoid building sub-linear regret algorithms.
This theorem provides a profound understanding of the challenges and encourages us to think more potential alternative model assumptions and solutions for optimizing $M^\natural$-concave functions in the online scenarios.

**Weaknesses:**

No obvious weaknesses.

**Questions:**

I am more curious about the conclusion in the bandit example.

1. In theorem 4.2, does $sReg_T$ is equivalent to the error in the Best-arm identification problem?

2. Does the bit-O term in theorem 4.2 and 4.3 hide the logarithmic term?

3. The theorem 4.3 claims that the proposed algorithm for stochastic bandit achieves cumulative regret as $O(K N^{1/2} T^{2/3}$. While according to [1], the tight lower bound of a consistent bandit algorithm is $\Theta(K^{1/2}T^{1/2})$. What kind of factor do you think causes the gap between your upper and lower bound?

[1] Auer, Peter, et al. "The nonstochastic multiarmed bandit problem." *SIAM journal on computing* 32.1 (2002): 48-77.

**Limitations:**

Please refer to Questions.

---

> ### Author Rebuttal · Authors · 2024-08-05
>
> We greatly appreciate the reviewer's insightful comments and positive evaluation. We address the questions below.
>
> > 1. In theorem 4.2, does $sReg_T$ is equivalent to the error in the Best-arm identification problem?
>
> Our ${\rm sReg}_T$ is essentially the same as the error measure in the best-arm identification, like the one used by Rejwan and Mansour [41]. To clarify the connection to Rejwan and Mansour [41] mentioned in the conclusion section, there is a bit of difference: they consider the $(\varepsilon, \delta)$-PAC guarantee, i.e., the high probability bound on the expected error, whereas our ${\rm sReg}_T$ focuses solely on the expected error. Nevertheless, their sample complexity lower bound [41, Theorem 10] of $T\gtrsim\Omega(N/\varepsilon^2)$ applies to any $K\le N/2$ and $\delta \in (0, 1/2)$, and hence Markov's inequality implies that our bound of ${\rm sReg}_T = O(K^{3/2}\sqrt{N/T})$ in Theorem 4.2 is tight with respect to $N$ and $T$ when $K=O(1)$.
>
> > 2. Does the bit-O term in theorem 4.2 and 4.3 hide the logarithmic term?
>
> The big-O notation in Theorems 4.2 and 4.3 does not hide any logarithmic terms. Our algorithms in Section 4 employ the minimax optimal strategy (MOSS) (e.g., Lattimore and Szepesvári [24, Chapter 9]), whose regret bound is free of $\log T$ factors that are typically present in the regret bounds of standard UCB-type algorithms. Consequently, our regret bounds do not involve logarithmic factors. To add further, this strategy could slightly improve previous explore-then-commit approaches for stochastic bandit submodular maximization [37, 38], which contain $\log T$ factors.
>
> > 3. The theorem 4.3 claims that the proposed algorithm for stochastic bandit achieves cumulative regret as $O(KN^{1/2}T^{2/3})$. While according to [1], the tight lower bound of a consistent bandit algorithm is $\Theta(K^{1/2}T^{1/2})$. What kind of factor do you think causes the gap between your upper and lower bound?
> >
> > [1] Auer, Peter, et al. "The nonstochastic multiarmed bandit problem." SIAM journal on computing 32.1 (2002): 48-77.
>
> We greatly value this insightful question. In our opinion, the primary challenge stems from the presence of exponentially many arms: even if we restrict the domain to $\\{0, 1\\}^V$, there are about $N^K$ arms. This leads to the following dilemma:
>
> 1. If we aim to achieve $O(\sqrt{T})$ regret, we may naively apply a UCB-type algorithm to the multi-armed bandit problem with about $N^K$ arms. However, this approach results in $O(\sqrt{N^K T})$ regret, and the algorithm requires exponential time per round in general.
> 2. Alternatively, we can design more efficient strategies based on offline algorithms. In our case, we have designed a no-regret strategy by executing a UCB-type algorithm in each iteration in the greedy algorithm, as described in Section 4. This method employs $K$ no-regret learners; however, only a single bandit feedback is available in each round. Due to this limited feedback, we need sufficient exploration across $K$ iterations in the greedy algorithm, as the explore-then-commit strategy does. Consequently, we incur $T^{2/3}$ regret as in Theorem 4.3.
>
> Furthermore, as detailed in the above [global response](https://openreview.net/forum?id=NnoAj91HZX&noteId=RjP8EjP159), a recent lower-bound result by Tajdini et al. (2023) suggests that we cannot do significantly better than the above two strategies in the case of submodular maximization. While there might be a slight possibility of achieving better regret by focusing on the $\text{M}^\natural$-concave case, we conjecture that we cannot achieve $\sqrt{T}$ regret that depends only polynomially on $K$ and $N$.
>
> We hope our responses have adequately addressed the reviewer's questions. Please do not hesitate to reach out during the discussion period if there are any further questions.

---

> > ### Comment · Reviewer_yBbo · 2024-08-13
> >
> > Thanks for your detailed explanation of the tightness of the $O(KN^{1/3}T^{2/3})$ regret bound in Theorem 4.3. Please add a formal justification in your paper based on your findings from Tajdini et al. (2023) if you can. I will not change my score since I have already supported this paper in my original review.

---

> ### Author Response · Authors · 2024-08-13
>
> We deeply appreciate the reviewer's continued support of our paper. We will make every effort to include a rigorous discussion on the tightness of our $O(KN^{1/3}T^{2/3})$-regret bound based on Tajdini et al. (2023).
>
> Just to be thorough, we would like to add to our previous response that the difficulty in achieving $O(\sqrt{T})$-regret arises not only from the exponentially many arms but also from the *non-linearity* of $\text{M}^\natural$-concave reward functions. While combinatorial bandits with linear rewards admit $O(\sqrt{T})$-regret algorithms such as COMBAND (Cesa-Bianchi & Lugosi, 2012), achieving $O(\sqrt{T})$-regret that depends polynomially on the problem size is significantly more challenging when dealing with non-linear rewards, as suggested by Tajdini et al. (2023) for the case of submodular rewards.

---

### Author Rebuttal · Authors · 2024-08-05

## **Global response: a discussion on the tightness of the $O(KN^{1/3}T^{2/3})$ regret bound in Theorem 4.3**
We sincerely thank all reviewers for their efforts in reviewing our paper and providing invaluable feedback. We are pleased to see that all reviewers have positively evaluated our work.

Upon reviewing the comments, we noted that multiple reviewers are curious about whether our  $O(KN^{1/3}T^{2/3})$ regret upper bound in Theorem 4.3 for stochastic bandit $\text{M}^\natural$-concave maximization is improvable. We revisited this issue and discovered an interesting recent preprint by Tajdini et al. (2023), titled "Minimax Optimal Submodular Optimization with Bandit Feedback." This paper studies stochastic bandit monotone submodular maximization with a ground set of size $N$ and a cardinality constraint of $K$. They showed that there is a lower bound of
$$ \Omega\left(\min_{i\le K}\ (K-i)N^{1/3}T^{2/3} + \sqrt{\binom{N-K}{i}T}\right) $$
on *robust greedy regret*, which intuitively compares the learner's actual reward with that of the output, denoted by $S_{\rm gr}$, of the greedy algorithm applied to the underlying true function. This lower bound implies that $O(KN^{1/3}T^{2/3})$ robust greedy regret is inevitable when $T$ is small in the submodular case; furthermore, the explore-then-commit strategy can achieve this regret bound. The $\sqrt{\binom{N-K}{i}T}$ term can be interpreted as the regret bound achieved by regrading all $\binom{N-K}{i}$ subsets as arms and using a UCB-type algorithm. Thus, the lower bound represents the best mix of the two regret terms achieved by the explore-then-commit strategy and the UCB applied to exponentially many arms.

Currently, we have confirmed that the proof for establishing the lower bound by Tajdini et al. (2023) does not directly apply to our stochastic bandit $\text{M}^\natural$-concave maximization problem. Specifically, the function used by Tajdini et al. (2023) for obtaining the lower bound is submodular but not $\text{M}^\natural$-concave. Nevertheless, the situations of Tajdini et al. (2023) and our problem in Section 4, with the domain restricted to $\\{0, 1\\}^V$, are remarkably similar:

1. Since the greedy algorithm applied to the unknown true $\text{M}^\natural$-concave function $f^*$ can find an optimal solution $x^*$, we have $x^* = S_{\rm gr}$ and hence the notion of robust greedy regret in Tajdini et al. (2023) essentially coincides with the standard regret in our case.
2. The explore-then-commit strategy and the UCB with exponentially many arms also achieve the $O(KN^{1/3}T^{2/3})$ and $O\left( \sqrt{\binom{N-K}{i}T} \right)$ regret bounds, respectively, in the $\text{M}^\natural$-concave case, where the former is the very result our Theorem 4.3 states.

Considering these facts, it is highly plausible that we can construct a hard instance of stochastic bandit $\text{M}^\natural$-concave maximization similar to Tajdini et al. (2023) to establish the same regret lower bound. Thus, we conjecture that our $O(KN^{1/3}T^{2/3})$ regret bound in Theorem 4.3 is tight in $K$, $N$, and $T$ if we want to avoid exponential factors, such as $N^K$, regardless of the value of $T$, which is a common desideratum in the context of combinatorial bandits. We will include an extensive discussion of this open problem in our revised manuscript.

We deeply thank dedicated reviewers, whose insightful comments have brought our attention to this interesting connection to Tajdini et al. (2023). We hope the information provided above helps reviewers understand that our $O(KN^{1/3}T^{2/3})$ regret bound in Theorem 4.3 may be close to being tight.

---

### Decision · Program_Chairs · 2024-09-25

**Decision:**

Accept (poster)

**Comment:**

This paper studied the online bandit problem of maximizing M-concave functions. Through a new error decomposition, it established regret guarantees in the stochastic setting. More noteworthy was the hardness of regret minimization in the adversarial case, which the reviewers also appreciated. Happy to recommend this paper for acceptance.